# Living and Researching the COVID-19 Pandemic: Autoethnographic Reflections from a Co-Research Team of Older People and Academics

**DOI:** 10.3390/ijerph21101329

**Published:** 2024-10-08

**Authors:** Louise McCabe, Tamara Brown, Roy Anderson, Liz Chrystall, David Curry, Margot Fairclough, Christine Ritchie, Pat Scrutton, Ann Smith, Elaine Douglas

**Affiliations:** 1Faculty of Social Sciences, University of Stirling, Stirling FK9 4LA, UKelaine.douglas@stir.ac.uk (E.D.); 2School of Health, Leeds Beckett University, Leeds LS1 3HE, UK

**Keywords:** co-production, co-research, older adults, COVID-19, autoethnographic

## Abstract

This article describes and reflects upon the work of a co-research team on the Healthy Ageing in Scotland (HAGIS) ‘COVID-19 Impact and Recovery’ study (January 2021 to November 2022). The co-research team (seven older adults and three academics) was constituted near the start of this project; the team contributed to the development of recruitment materials and research tools and undertook qualitative research and analysis with older adults living across Scotland. This article provides a collaborative autoethnography about the activities undertaken by the team, the impact of the co-research process on the individuals involved, and the research findings and reflects the realities of co-research during the COVID-19 pandemic. Team members describe benefits, including increased confidence, new skills, and social connections, and reflect on the increased validity of the findings through their close involvement in the co-creation of knowledge. The process of team building and the adoption of an ‘ethics of care’ in our practice underpinned the success of this project and the sustainability of the group during and after the challenging circumstances of the pandemic.

## 1. Introduction

The Healthy Ageing in Scotland (HAGIS) COVID-19 Impact and Recovery study aimed to understand the experiences of people aged over fifty living in Scotland during the COVID-19 pandemic. It was a large-scale mixed-methods project that included scale development (worries emerging from the COVID-19 Pandemic), a survey (online, telephone, postal), an eDelphi exercise, and qualitative interviews [1]. Seven co-researchers worked as volunteers alongside the academic team. We use the term ‘co-researchers’ to refer to members of the research team who worked as volunteers and were drawn from the participation population for the research project. Co-researchers did not have a background in research. Co-researchers played an active role in developing recruitment materials, drafting research questions and research tools, undertaking data collection through interviews and focus groups, qualitative data analysis, and presenting and publishing project findings.

This article explores how working with and as co-researchers is a mutually beneficial process. We share our learning from our experiences of co-research with older people who relied on distance-based participatory methods during the pandemic. We present what we did as a team and how we worked together and reflect on the impact of this process on the research findings. We reflect on how co-production is not only a knowledge-making endeavor but also a social space in which knowledge is produced and shaped by personal experiences and social relations. We discuss how the co-research team was built and sustained, and how we strived for an ethical approach to working together during the particular challenges of the pandemic.

This article first provides descriptive information about the work of the co-research team to give a clear context for the findings and discussions that follow. The ‘findings’ part of this article presents a collaborative autoethnographic approach to understanding the process and experience of co-research. This is based on transcripts from reflective conversations during team check-ins and from a set of individual written responses provided by each team member. The article was co-written by nine members of the co-research team: two academic researchers and seven co-researchers.

### Background

Co-production and the work of co-researchers are increasingly prioritised in gerontological research and there are a growing number of examples of co-research teams doing research ‘with’ older people rather than ‘about’ older people [2]. The approach to co-research presented here builds on a long history of participation in research that has evolved from people participating as research subjects or sitting on advisory panels to approaches where members of the community of interest become directly involved in shaping, undertaking, and disseminating research projects [3]. James and Buffel [2], in their review of co-research with older people, note that gerontology was late to adopt these approaches, but over the past twenty years, there has been a steady increase in the involvement of older people. Co-production of research activities and co-creation of knowledge are now seen as essential elements of research, particularly within health and social science [4].

There are different approaches and a number of terms to describe these activities such as co-production, participatory research, peer research, and co-research, all used in different ways and with different meanings adopted in different settings [5]. Co-research describes more active involvement by people with lived experience in the research process itself [2,6]. As part of a wider field of participatory approaches, co-research aims to break down barriers between researchers and those being researched, promoting equity and exchange to produce more meaningful outcomes [7,8]. Co-production and co-research should promote an ethos of meaningful participation and shared work, addressing power imbalances between those undertaking research and those who are the subject of research [9,10]. Further, participatory approaches can help reduce the gap between academic interpretations and lived experience, enabling the co-creation of knowledge about those being researched [11].

In co-research, there is a balance between ensuring that people’s participation is meaningful while also ensuring ethical approaches that safeguard their well-being within the research process [12]. Undertaking an egalitarian approach to co-research may place significant demands on volunteer co-researchers; thus, it is vital that academics carefully consider the ethics of care when undertaking co-research [3,5,9]. Balancing hierarchies, building trusting relationships, sharing control, and ownership of research processes are highlighted as common themes in reviews of ethical practice in participatory research [5] (p. 287). Taking an ‘ethics of care’ approach involves a focus on relationships and caring, recognizing that research is a human endeavor that involves individuals working together [9].

Reviews of participatory methods adopted and refined during the pandemic provide evidence of a wide range of online and creative approaches that were successfully developed for working with research participants at a distance [13,14]. Examples of creative methods include photovoice to increase meaningful participation, online video focus groups to enable wider participation, video diaries, creative methods, and collaborative autoethnography (CAE) to allow for collective reflection [13]. Sattler et al. conclude that a hybrid approach of online and face-to-face participatory research is the way forward for post-pandemic research [14].

This article presents a collaborative autoethnography [15] about the collective experiences of undertaking participatory research with a reliance on distance-based participatory methods during the COVID-19 pandemic. We aim to inform practice and process in gerontological social science research, including recommendations for ethical practice in co-research.

## 2. Materials and Methods

Autoethnography is a reflective mode of reporting on one’s own experience and traditionally reports from a single perspective; in contrast, CAE allows for reflection within a group with shared experiences. It is defined as ’a qualitative research method in which researchers work in community to collect their autobiographical materials to analyze and interpret their data collectively’ [15] (p. 23). CAE has been shown to be an effective tool for illuminating co-research experiences and processes [16,17]. CAE follows an iterative process of individual writing, collective discussion and reflection, and group writing. The approach does not specify particular research methods, but it should encompass forums and processes that the group is comfortable with [18].

The idea for the CAE project emerged from regular team ‘check-ins’ where we reflected on our work, how people were feeling, and what our next steps might be. As such, we regularly spoke as a group about the co-research process and the impact on members of the team. It seemed a natural step to move from those informal team discussions to undertake the CAE. Here we worked through the steps using a combination of online reflective workshops, individual writing, paired writing, and group writing. These were all activities that the group had experience of through working on the HAGIS research project. CAE supports power sharing among a group and community building [5,15], and the process of writing this article mirrored the approach we took to co-research, supporting reciprocity as a form of blurring hierarchy.

The first step was to generate a series of questions to enable us to frame our input to the process, which we did during a guided reflective discussion in an online meeting. These questions were as follows:What did you bring to the table and how did that contribute to the work of the group?What training and skills development did you need/receive?What was the impact for you from being involved in the project?What was the impact on the research of having co-researchers involved?What ethical concerns were/are raised by working with co-researchers?

Each team member contributed written responses to the five questions. Nine written responses were received. The team collated responses and a pair of people from the team looked at each set of responses. These pairs wrote an initial draft to draw out themes within each response, which were shared across the whole team. The write-ups were discussed at three online meetings, the final one of which was recorded. The recorded transcript was added to the written responses to form another data source. Taking on board reflections from these discussions, each pair revised the writing about their theme, and these sections were brought together as a whole. Some pairs included an academic, but several were made up of two co-researchers.

The team then met in person for a one-day writing workshop where we reviewed the manuscript, refined the themes, and worked on the content of the discussion section of the article. This process led to the emergence of five key themes:Skills, enthusiasm, and empathy;Connection, confidence, and learning;Enhancing process, findings, and dissemination;Being ethical in co-research;Building and sustaining a co-research team.

The two academic members then edited the existing manuscript and wrote the discussion section, producing a first draft. This manuscript was then reviewed and edited by the team, drawing on skills in proofreading and grammar, including one member who was particularly skilled at summarizing ideas.

The work of the co-research team and the research methods for the wider project were approved by the General University Ethics Panel at the authors’ university (number: 485). Advice was sought from the panel regarding the collaborative autoethnographic process, and we were advised that no further ethical review was required. All authors of the article (except for the final author who is the principal investigator) are also participants in the CAE project; we discussed the implications of this for our anonymity and everyone provided consent for their data to be included in the article. In the findings, quotes are identified with the author’s initial and an A to denote an academic researcher and a C for co-researcher. A summary of the team is provided in Table 1.

### 2.1. The Work of the Co-Research Team

To understand the reflections offered by the team members, we start by describing the work of the co-research team and the key activities undertaken. The context of the pandemic and constantly changing restrictions impacted both the methodology of participation and our lived experience.

The co-researchers were recruited at an early stage of project delivery (within four months of funding being awarded); this meant they were not involved in the design of the project but came in at an early enough stage to influence the design of participant recruitment materials and research tools.

#### 2.1.1. Recruitment and Kick-Off

The project aimed to recruit 12 co-researchers aged over 50 and living in Scotland to reflect the participant population for the study. A recruitment flyer was developed and disseminated through professional and community networks and the call for co-researchers was included in communication activities at the time of the project launch. Over a month, we recruited eight people to the team, of which seven are co-authors. One person dropped out of the process early on—this person was the only one who was still in employment. Of the seven people, two answered an advertisement in a local voluntary organisation’s bulletin; one attended a meeting with Voluntary Health Scotland where the HAGIS project lead was speaking; two read about the HAGIS project in the newspaper; and two received an email from other organisations where they had expressed an interest in research. All co-researchers live in East and Northeast Scotland.

Restrictions relating to the COVID-19 pandemic meant that the bulk of the co-production work was carried out remotely. The team worked wholly online for the first six months, with subsequent additional infrequent in-person meetings. The seven co-researchers were able to engage with the technology and we supported each other to learn about the functionality of Microsoft Teams. Equipment such as headphones was provided (paid for out of project funds) to members of the team who needed these. Online working could have been a potential barrier for prospective co-researchers.

#### 2.1.2. Getting Started

The team quickly established a pattern of monthly meetings, with brief notes and actions subsequently provided by an academic team member to enable those not attending to stay up to date. Meetings were usually planned for an hour but often lasted longer due to lively discussions and providing time to catch up with each other. Starting in September 2021, we were able to hold in-person meetings in line with COVID-19 restrictions. At the time of writing, we held five in-person meetings in Dundee, Perth, and Stirling; the final one in Perth was organised by the co-researchers. Meetings followed a set agenda agreed by the team but always provided space for discussion and reflection on emerging topics and allowed time for social interaction and getting to know one another. The meetings provided a space for collaboration and professional friendship, nurtured by a deep sense of collegiality amongst the group.

The first activity the group became involved with was to assist in the design of a scale to measure fear of COVID-19 as part of a wider project. This involved a ‘think aloud’ exercise [19], conducted on a one-to-one basis, to help design an assessment tool. Group discussions followed to design recruitment documentation and survey names and to review and offer feedback on survey questions.

#### 2.1.3. Qualitative Fieldwork

Following this, the team started work on the qualitative aspect of the project, and this comprised the bulk of the team’s work going forward. This work involved recruitment, interview schedule design, data collection, analysis, and write-up. The co-researchers supported participant recruitment by circulating the recruitment flyer in their communities and voluntary and professional networks. The co-researchers were closely involved in designing the interview schedule and developing clear and engaging questions that were sensitive and relevant. The interview schedule covered general feelings and thoughts about COVID-19 and the restrictions imposed, social connectedness, health, finance, work, and technology. The interviews were semi-structured [20], with set questions and prompts to provide some consistency across the project. This ensured we had data to contribute to the mixed-methods aspect of the wider project as well as the scope for interviewers to probe more deeply into a topic or theme of relevance to the participant. This enabled an inductive interpretivist approach to our data analysis.

Data collection involved individual and small-group interviews, mostly conducted online. A small number of one-to-one interviews were conducted on the telephone and face-to-face, once restrictions were lifted and according to a participant’s preference. Most interviews were facilitated by one co-researcher and one academic researcher; five interviews were conducted only by an academic interviewer as none of the co-research team were available. The academic researcher took the lead on introductions and completing consent processes while the co-researcher took the lead on the interview questions and prompts. Academic researchers only interjected when curious to probe more about a topic.

All interviews included a 30-min pre-meeting and a post-interview debrief between the academic and co-researcher. This meant co-researchers could ask questions and have time to chat in the pre-meeting to ready themselves for an interview. During the debrief, academic researchers could provide additional support if interviews had been distressing. Each co-researcher completed a “Facts, Feelings and Reflections” review immediately after each interview to record what the interviewer learnt and felt during the interview and their reflections on these. This was a format used by one of the co-researchers during her counselling training and practice and the group agreed it would be a helpful tool for them on this project.

Co-researchers were also involved in data analysis, a challenging task but one that the group learnt from and enjoyed (in the end). The analysis involved a review of anonymised transcripts to develop a coding framework and then an analysis of collated data ‘nodes’ on different topics. Analysis was conducted individually and then discussed during an online group meeting, with an in-depth analysis taking place in an in-person group meeting.

#### 2.1.4. Training

To support the co-researchers with the necessary knowledge and skills for data collection and analysis, a series of group training sessions were held. The four online sessions covered the following topics:

Introduction to Research Skills: considered ‘What is, Why and How’ we conduct research, considering our own experiences, social research methods, and specific methods used in the HAGIS project (August 2021).

Research Skills II: considered the three “C’s”–Consent, Capacity, and Confidentiality. This session reflected on the meaning of these terms and their relevance in qualitative research and linked to legislation such as the Adults with Incapacity (Scotland) Act 2000 and the Data Protection Act 2018. (September 2021)

Research Skills III: focused on interviewing skills, thinking about listening skills, and how to use prompts and cues in an interview setting. This session included a listening skills test, which was excellently led by a co-researcher who is a qualified counsellor. This brought out the serious side of truly listening while also providing a little laughter. A subsequent online session was used to role-play interview scenarios with one of the academic staff taking on the role of a tricky interviewee (October 2021).

Interview skills training was covered in more depth at an in-person workshop held in Dundee. At this workshop, we considered different interview scenarios, such as the participant becoming very emotional or interviewees bringing up sensitive topics or revealing potential harm to themselves or others. This session also provided an opportunity to role-play some interview scenarios.

Research Skills IV: focused on qualitative analysis, which was covered in three stages: data analysis, thematic analysis, and familiarisation with the data [21]. We found themes within transcripts and discussed key emerging topics, noting similarities and differences between topics. We arranged the topics under broader headings from our interview guide (February 2022).

#### 2.1.5. Dissemination Activities

Key themes were then presented at two online and one in-person conference, which were important activities for the team. Minor crises of confidence were expertly dealt with by the academic researchers. Formats were agreed upon that enabled every co-researcher to contribute in a professional manner and one that was comfortable for them. For example, some members preferred to pre-record their contributions, some read from a script, and others spoke to PowerPoint slides. Feedback from all three conferences was extremely positive.

## 3. Results

The sections that follow present reflections from the team on different aspects of their experiences as members of the co-research team. These reflections provide important learning on what can be achieved through co-research, the benefits co-research brings to those involved and to the quality of research, and the consideration of ethical practice when working in a co-research team.

### 3.1. Skills, Enthusiasm, and Empathy

Members of the team brought a mix of personal qualities, lived experience, skills, and knowledge that contributed to the work on the project. Co-researchers emphasised their insider status; as people drawn from the same population as the project participants, they share characteristics such as age and generation, and in addition to this, they bring similar life experiences. Co-researchers stressed that this gave them empathy and sensitivity to the experiences of participants and, sometimes, a shared language in which they were able to talk about their lives. Further, it made them interested to engage with and learn about the experiences of the participants.

*Personally, I felt I brought age, experience, empathy. age, was a benefit, I felt it helped the interviewees, all over 50, to relate to me. Empathy, having gone through mental anguish, OCD* [referring to the diagnosis of obsessive-compulsive disorder] *which during COVID returned, made me aware of, and could relate to the interviewees.*(AS-C)

Some co-researchers had specific experiences during the pandemic that helped shape the data analysis and draw out important findings, such as their roles as unpaid carers and one person’s experience as a ‘shielder’ during the pandemic. We use the term ‘shielder’ to refer to someone who was advised by the NHS to abide by a stricter set of restrictions during the COVID-19 pandemic due to a higher risk of mortality due to COVID-19.


*I was a shielder throughout the COVID pandemic which gave me an understanding of some of the issues which arose for other shielders, including an acute sense of loneliness and isolation, and enhanced levels of fear for myself and the person I care for. The heightened sense of anxiety I underwent lent me great empathy with participants of the research.*
(MF-C)

Co-researchers brought enthusiasm and a willingness to learn and gain new skills and experiences. Academic researchers also had life experiences that enabled them to connect with participants through their own experiences during the pandemic.


*I am female, aged 50, with four children (three still at home) and I live in Perthshire, Scotland. I brought both academic skills to the table but also ‘softer’ skills such as empathy. I am used to working within teams from different disciplines and I like to think that I am approachable as well as organised and reliable. I contributed to a non-hierarchical working relationship within the team and took the time to get to know team members. I feel that we were on a journey together and grew as a team as trust was built.*
(TB-A)

Co-researchers also brought a broad range of skills and knowledge acquired during their careers or later in life as volunteers. These were usefully applied within the research project. These included expert counselling skills, knowledge of data processing, written and spoken communication skills, interview skills, management and teamwork, and knowledge of ethical issues and confidentiality.

*I brought both lived and life experience as well as counselling skills. As a lover of the English language, I could contribute to the clear content of questionnaires, reports* etc.(LC-C)


*Extensive and varied experience from a range of employment in public, private and third sectors. Excellent communication skills, both verbal and written, but particularly the latter. Presentation skills honed over the years. Experience of interviewing, both in work and community settings. Knowledge of subject matter as went through a nasty bout of COVID near start of pandemic.*
(RA-C)

Academic researchers also brought their own knowledge and skills of qualitative research, ethics, data collection, and analysis, that complemented the knowledge and skills of the co-researchers. This was shared with the team through the training described above and more informally within workshops and meetings.

### 3.2. Connection, Confidence, and Learning

This co-research experience led to new social connections and learning; it provided opportunities to contribute and find purpose and, at times, it was emotionally impactful. For academics, there were additional pressures in supporting members of the group who were experiencing their own distress and ill health during the pandemic. For example, when we met in person there were specific stresses about keeping team members safe within pandemic guidelines. The context of the pandemic meant that co-researchers were already experiencing challenging circumstances, which further heightened the emotional experience for the team. All co-researchers were able to express empathy with the experiences of the participants they interviewed.

On a social level, it enabled co-researchers to meet new like-minded people of a similar age. Over time, as meetings online continued and face-to-face meetings took place, acquaintances have developed into firm friendships. This led to increased contact with regular project meetings.


*It gave me a purpose, insight and a tremendous experience, made new working friends through working with a very good team of people with similar and different backgrounds.*
(CR-C)

For some, the way that the team gelled together was a surprise but confirmed their belief in the basic goodness of most people.


*I have found the experience very interesting and satisfying. I was surprised that a group of seven volunteers from different backgrounds would be able to “gel” and form a team as quickly as we did. This confirmed a belief that most folk are good.*
(DC-C)

Learning also had a significant impact on the team throughout the research. For some, it enabled them to stay active in an intellectual and productive way, particularly those who recently retired.


*It gave a whole new meaning to life during a time when it would have been very easy to become insular. It got my brain cells working again and meeting new people virtually, working with them and, eventually, meeting them in person to achieve a common goal gave my life real purpose.*
(LC-C)

Perhaps the most notable skill acquired during the project was how to conduct qualitative analysis. None of the co-researchers had any previous experience of doing this, so it presented a steep learning curve for everyone. However, all agreed it represented a useful addition to their skill set. For some, there was an element of surprise about being able to learn new skills ‘at their age’.

The research also presented opportunities for members of the team to rekindle skills, which they had employed frequently in the past–including interviewing, counselling, and networking. With regard to interviewing skills, some felt they had learned to become more reflective and sensitive to the needs of interviewees during interviews.


*Just having retired this was opportunity to continue to utilise previous skills and foster a (semi) active mind!*
(RA-C)


*I could rekindle old skill and develop new ones. A really good and interesting learning experience.*
(CR-C)

Learning was also an important outcome for academic researchers on the team.


*This project has helped me develop a more reflective and sensitive approach to doing interviews and focus groups and has enabled me to connect more emotionally with research participants and within the data analysis process.*
(LM-A)

As already noted, many team members felt they were making an important contribution to other people’s lives. Co-researchers were focused throughout on how the research findings could be most effectively disseminated to ensure impact. However, some frustration was felt during the dissemination phase when it became clear that there was no clear route to impact the project.


*The impact for me of being involved gave me a positive feeling that I was able to contribute to such a worthwhile project.*
(AS-C)


*Opportunity to make a difference to lives of others through dissemination of the key outcomes of the research.*
(RA-C)

Due to the regular points of contact via meetings, virtual as well as in-person, one academic team member regarded her time as a co-researcher as a much-needed connection to the outside world during the pandemic restrictions. Co-researchers appreciated the way in which the academic researchers built in opportunities for the whole team to spend time together, in person where possible.

Involvement in the project also provided team members with gratitude and a more positive perspective on their own experiences, in the wake of hearing about so many challenging stories from the interviewees of their experiences during the pandemic.


*The project experience has made me realise just how lucky I am especially after listening to the stories and experiences of some of the folk I interviewed.*
(DC-C)

Researchers grew in awareness of sharing the same feelings as others, brought about by the pandemic, and found comfort in that.


*I was not alone with my fears of the pandemic.*
(AS-C)

One member of the team, who had shielded and, thus, experienced much longer periods of enforced isolation than the wider public, commented on experiencing extreme feelings of anxiety and loss of confidence, which were actively alleviated through the sensitivity and understanding of other members of the research group.


*This sense of sharing feelings was particularly important given that the isolation brought about through the various lockdowns fractured the usual (traditional?) sense of security felt by many through engagement in community activities.*
(MF-C)

Others described positive feelings of enjoyment at getting to know and working with colleagues on the research project and becoming part of a team.


*I very much enjoyed getting to know and working with both my fellow co-researchers and our academic partners. There was a very real sense of working as a team.*
(PS-C)

For one co-researcher, her experience on the team was transformative in several ways but, importantly, helped her rediscover the enthusiasm and self-worth to pursue a degree course.


*I had also given up on completing my first degree. After attending my first face to face group meeting and conference, I was inspired to continue my degree. I have now submitted my dissertation which I hope will lead to my graduation, and without hyperbole feel that I have returned from the dead.*
(MF-C)

### 3.3. Enhancing Process, Findings, and Dissemination

Working with co-researchers from an early stage in the project (four months after funding was awarded) had a significant impact on the research. All the co-researchers focused their reflections on the benefits they brought to the interviews. Co-researchers helped shape the recruitment literature, ensuring that the language was accessible and attractive to the target audience, and brought a positive dynamic to conducting interviews, undertaking analysis, and disseminating findings.

Co-researchers impacted the quality and depth of interaction with the interviewees through their position as peers with shared characteristics and experiences.


*Having co-researchers involved working with the general public alongside the academics I feel, made it ‘real’ for the interviewees, made them more relaxed. The co-researchers being in the same ‘age’ group as the interviewees allowed them to feel comfortable along with the empathy the co-researcher brought. Answering the questions was more natural, more of a friendly chat, that brought out more meaningful responses.*
(AS-C)

The co-researchers were in the same broad age group as the interview participants, and this allowed the interview participants to feel comfortable and be more responsive and willing to share sometimes very personal experiences. It meant the interviews became conversational, with co-researchers listening and responding where necessary. Co-researchers were able to tease out further information from the participants by encouraging them to expand on their experiences.


*It is sensitivity to the issues and experiences of participants that makes co-researchers and experts by experience so uniquely invaluable in eliciting and capturing the thick description that enriches the data.*
(MF-C)

Co-researchers brought different and varied perspectives to the research, based on their own life and employment experiences. This meant that co-researchers were able to keep the research process grounded in real life and provided a counterbalance to the more academic approach of experienced researchers.


*Having co-researchers involved in this research has, I believe, strengthened the validity of the findings. Having direct involvement of the co-researchers in the data collection and analysis ensured that we attended to the more important issues and themes within the data. Their reflections on the data analysis ensured we headlined what was important to the participant group.*
(LM-A)

Co-researchers also brought emotion to the interview interactions through shared experiences. This deepened the engagement with participants and the depth of data produced.


*Perhaps one of the things we are bringing is the emotion, that fact that we are living it, we’re not just talking in dry terms about an issue that is ‘out there’. You know, this is affecting us as we speak.*
(MF-C)

Having co-researchers lead with the questions during the interviews brought added benefit for the academic co-researchers.


*Having co-researchers lead with the questions during the interviews allowed the full-time researchers to step back and listen, thereafter engaging when they felt appropriate.*
(DC-C)

The quality and impact of conference presentations were significantly improved through the involvement of co-researchers. The evidence that the co-researchers had taken their responsibilities very seriously and had been fully involved was apparent when opportunities arose to present the findings. The whole team presented the interim findings at three international conferences [22,23,24]. Then, at the end of the project, the co-researchers gave two presentations at a HAGIS report launch event at the University of Stirling, involving academics working on other longitudinal studies of ageing. Presenting at a conference was a new experience for most of the co-researchers and some preferred to pre-record their contributions rather than present live. The team supported each other to practice and gain confidence. The first presentation was on the findings, and the confidence they showed not only in presenting the findings but in fielding questions was evidence that their role within the team had been neither ‘window-dressing’ nor tokenistic. Co-researchers’ own experiences helped highlight particular experiences during the pandemic, especially one co-researcher’s experience of shielding.


*When MF was answering questions at the final conference, I was really moved by the way she presented and answered the questions; it definitely got the heartfelt message over.*
(CR-C)

### 3.4. Ethical Practice in Co-Research

Working in a co-research team raises several ethical considerations related both to the well-being of the co-researchers themselves and to the integrity of the research processes undertaken. It is important that academic researchers accept a duty of care towards the co-researchers on their team and ensure that risks are mitigated [9].

It was recognised that co-researchers were giving a lot of their time and effort to the project on a voluntary basis. As co-researchers became more involved with the research process, the academic co-researchers were careful to emphasise the voluntary nature of the work. Academic co-researchers always checked explicitly with each team member about their wish to take on different pieces of work. There is always a risk in such a team, of an imbalance of power (and knowledge) between the paid professional researchers and the volunteer researchers [10]. There was a potential for loss of trust, but this did not materialise, probably due to the ground rules being made explicit at the outset, resulting in the co-research team being viewed as a team of equals.


*For me, the main ethical issue is the potential power imbalance between skilled and paid academic staff and volunteer co-researchers. It was helpful to have this explicitly raised at the beginning of our involvement. Once we started working together it was clear that this would not actually present a barrier, and we worked together very well as a team, with our voices, opinions and ideas listened to, heard and respected.*
(PS-C)

Co-researchers reflected on the notion of payment, and there was consensus that payment for their time would have introduced an unwanted pressure; volunteering (without payment) meant they could contribute as little or as much as they wanted and when they wanted. The co-researchers were attracted to joining the project because it was clear that it was voluntary; they were highly motivated to join and found other, non-monetary rewards from engaging with the process as we have discussed.

While the involvement of co-researchers adds strength and rigour to the research, it also raises potential risks for co-researchers during challenging interviews with participants. There was a risk that co-researchers would identify with issues raised in interviews and potentially suffer emotional distress. The impact of seeing and hearing difficult stories from some of the participants had the potential to deeply affect co-researchers, especially if they were conducting the work from their homes and in isolation.


*As we co-researchers could closely identify with at least some of the issues faced by participants, there was an increased risk that we might be deeply affected by their concerns and potential distress.*
(MF-C)

We took steps to mitigate this risk, including working in pairs (one academic researcher and one co-researcher); making time for a pre-interview planning meeting and post-interview debrief; and encouraging co-researchers to write up their reflections after each interview. In fact, several interviews did raise strong emotions for participants and co-researchers, and these support processes were found to be effective in ensuring all co-researchers were able to reflect on and process their emotions.


*I think the fact that we all seemed to gel. And we really felt we were a team. To me it was very important.*
(CR-C)

Co-research methods might also raise concerns that co-researchers fail to follow standard ethical guidelines and practices, and consequently, the research outcomes are compromised. These were addressed through the co-interview process described above and the provision of training. Co-researchers also brought their own knowledge of concepts such as confidentiality and the General Data Protection Regulation (GDPR) from their work and other volunteer roles, which they were able to share with the group.


*The academics made sure that we were knowledgeable and up to date whilst working with the public, for instance informed consent, integrity, the need for privacy and anonymity, confidentiality, voluntary participation.*
(AS-C)

Additionally, the interviews were led by co-researchers with the academic researcher able to listen and intervene if needed; however, in practice, this was seldom found to be necessary.

Working with co-researchers did place additional responsibility on the academic researchers, with the time and effort needed to train and support the group on an ongoing basis.


*The same skills of empathy, tolerance, patience and positive regard that were sought in the co-researchers were required by the facilitators towards us. The facilitators also accepted an additional burden of work in preparing and exploring issues with us, and timetabling preparation and workloads (herding cats?).*
(MF-C)

This responsibility was exacerbated by the pandemic due to the need to work online and when we were able to meet in person, to address the ongoing risks of COVID-19 infection.


*There was a degree of ‘emotional labour’/investment that was probably higher than I expected when I started this project–working with the co-researchers was immensely enriching to myself and to the project but this was balanced at times with the extra work and sometimes stress of supporting the group.*
(LM-A)

Working as a co-research team necessitated careful reflection on ethics and on the duty of care that the academics have towards the co-researchers, and for the team to continually reflect on their work and well-being at different stages of the project.

### 3.5. Building and Sustaining the Team

The academic team led through facilitation and took considerable time to invest in the needs of the team to build a cohesive team. On reflection, a ‘safe space’ was crucial to building this relationship. As one co-researcher put it: “we were able to disagree, but we didn’t fall out”. Everyone was valued for their contributions to the group, and we were able to challenge each other in a productive way. An important component of the team was its non-hierarchical structure, which promoted a balance of the values of professionalism and lived experience.

There was a willingness to open up to each other and work at a more emotional level and make connections with each other. We lost the “professional veneer” (MF) and were able to achieve that due to the trust that we built among the group.

The development of relationships among team members was challenged by the online methods we were using, and when we were able to come together in person, it was notable that our connections with one another were strengthened. For one participant who was shielding, it was much later in the project when she was able to meet everyone else in person for the first time and she reflected that she felt left behind because of this.

When the project officially ended, all members of the team decided to continue to work together to write this article and other articles reporting the findings from the project. The connections across the team are illustrated in this summary:


*We came together as seven individuals and have become a friendly team as we have got to know more about each other. Two members of the team have family in America and have been able to share the trials and tribulations of not being able to travel to see them, then the problems faced when flights started up again. The same two co-researchers have very similar tastes in music, discussed at some length on the journey from the train station to the University. Three of us found a shared liking for whisky and an invitation to meet at the Whisky Centre in Edinburgh has yet to be fulfilled. We all learned a great deal from one member who had a very traumatic experience in relation to shielding. She revealed details of which we may have been otherwise unaware, and this gave added depth to our presentations–there were a few tears.*
(LC-C)

## 4. Discussion

Co-researcher involvement positively impacted throughout all the stages of the project. Co-researcher demographics and life experiences were like those of the participants, which meant that co-researcher involvement kept the research real and meaningful. Co-researchers brought diverse skills and experiences to the team and were able to encourage engagement and dialogue during the interviews. Co-researchers kept the focus on the important issues and themes that emerged from the data, and their reflections enriched the analyses. Co-researchers strengthened the validity of the findings and were able to headline the most important findings through authentic reflection on the data, which resonated with their audience and furthered impactful dissemination with stakeholders, reflecting the findings from the review by James and Buffel [2]. When considering other co-production research, there are fewer examples of co-research approaches being applied to data analysis and writing, but there is a clear trend towards this [5,16,25], to which our work contributes.

The co-research process led to positive outcomes for both academic and volunteer co-researchers. For the co-researchers, the experience provided new skills and knowledge as well as growing self-confidence and the opportunity for new experiences. Gaining research skills, taking part in conferences, engaging with new technologies, and writing for different audiences were all tangible skills-related outcomes for the co-researchers. The team also valued the relationships built with members of the team. These benefits of participatory research and co-research are reported more widely with notions of joy and satisfaction from the learning that comes with being part of research [2,8].

Brown [3] offers a useful continuum of participatory research from PPI through to co-research, illustrating the increasing involvement of people with lived experience as part of research teams over time. However, she notes that few research studies reach the most egalitarian approach to research, with most falling somewhere below that [19]. Our work sits towards the more active end of the spectrum, with scope for future co-production beginning at the research development (pre-funding) stage in line with recommendations from Cotterell and Buffel [26]. Earlier input into the design of survey questions and recruitment documents for the HAGIS project would have been valued by the co-researchers. The presentation of our research findings by the co-researchers at various conferences demonstrated the emancipatory nature of the experience for the co-researchers.

Reflection and thought are required for the practical and emotional support co-researchers may need, as well as for the delicate power balance between academics and co-researchers in the production of knowledge within research. Ozkul [10] argues that there is still a long way to go to ensure that power balances are addressed in meaningful ways and to involve people with lived experience throughout the research process. She argues for greater participation in formulating research aims and in how we conduct research. Having this notion of power on the table from the start helped us explicitly reflect on and address power imbalance in the group.

The work of the team described and discussed here was relational in nature, and thus, it was important to consider the ethics of working with co-researchers and ensuring their active role in the research. Here, this was accomplished through adopting an approach in line with Tronto’s [27] model of ‘ethic of care’ that emphasises the importance of caring for and caring about members of a co-production team and being cognizant of relationships, emotions, and potential power imbalance within teams [9,27]. Tronto [27] proposed an ‘ethic of care’ model that endeavors to explain and examine the concept of care and that enables academics to interrogate ideas of care within different contexts. Core to this model are concepts of power, relationship, and reciprocity. This model has been applied within participatory research teams to examine the relationality and care that exists between members of co-research teams [5,16,28]. Within our group, we cared for and cared about the research process and outcomes and supported reciprocity amongst team members. We maintained and encouraged a fundamental respect for each other and our individual strengths. Further, the group shared a desire to learn from their own and others’ experiences during the pandemic and to share this learning widely. We cared for and cared about each other; we allowed time to build relationships; took an interest in each other that went beyond the immediate work of the project; recognised the emotional aspect of our work; and through this, were able to offer support when people were going through difficult times.

Uniquely, this article highlights lessons learned from the process of conducting co-research with older people during a pandemic. This article adds to the documentation of co-production activity involving older people and continues to build the evidence base through critical appraisal of the co-production process [26], highlighting shared values and developing principles of co-production strategy [29]. Further, the context of the pandemic provided the impetus to develop ways of working together in an ethical manner but at a distance, extending the evidence about participatory methods adopted during the pandemic [13,14]. The application of the CAE approach to writing this article enabled us as a team to take forward the ethos and ways of working developed during the original research project into writing this article, demonstrating the value of CAE in supporting reciprocity and addressing power imbalances.

### Limitations to Our Approach

The limitations to the CAE project presented here reflect the limitations in our wider work as a co-research team. All co-researchers self-nominated themselves to be part of the team; all are retired and undertake other voluntary work. As such, there were shared characteristics across the group. This brings potential limitations to the transferability of this co-production work to groups from minority and marginalised communities and to the ability of the team to engage empathically with a diverse range of participants. For future work, it would be beneficial to recruit a more diverse group of co-researchers. Further, the fact that all co-researchers were also involved in other voluntary activities suggests they already saw value in volunteering and were more likely to put themselves forward to undertake co-research on a voluntary basis. This may not be true for people in employment or who have caring responsibilities or from more marginalised groups of older people. Again, this highlights the specific nature of our co-research team and the wider limitations of working with volunteers. There is a potential that co-research with older people who have the time and resources to take part creates further inequalities [30].

For this project, the experience of working during pandemic restrictions impacted the process and outputs of the research. Our hybrid approach of online and in-person meetings worked well, and online meetings may have even increased engagement for some co-researchers by enabling them to join meetings even when on holiday. However, we are aware of the risk of digital exclusion and the importance of having some in-person meetings to build relationships within the team. One co-researcher reflected on the negative impact of missing out on the initial in-person meeting, and we all acknowledged the important investment of these in-person meetings for team cohesion. Some people’s voices were difficult to hear online, especially soft or high-pitched voices, and this was difficult to talk about for fear of causing offence; online working did not enable a ‘quiet corner’ to discuss this.

Finally, we acknowledge that the co-research approach would have been strengthened with the involvement of co-researchers in the conception and bid-writing phases of the project.

## 5. Conclusions

We advocate for a co-research approach and highlight, through the examples shared here, the value of co-research and the potential benefits of co-research during challenging experiences such as a pandemic. Reflecting on our experience of co-research, we recognise the hard work contributed by all members of the team but also the rewards felt by all members and the positive impact on the quality and ethical approach adopted during this research. Co-research provided an opportunity for the older team members to feel they were making a meaningful contribution and creating new social connections at a time when people were physically isolated. It is important to recognise that co-research will, by its nature, be relational and emotional and to ensure all team members have access to support. Going forward, we recognise the importance of including co-researchers from the planning stages of any research onwards and, on reflection, wish that we had all started a reflective diary early on to track the process of co-research.

## Figures and Tables

**Table 1 ijerph-21-01329-t001:** CAE participants.

**Gender**	Female (7)	Male (2)			
**Age**	40–49 (1)	50–59 (1)	60–69 (2)	70–79 (5)	
**Ethnicity**	White Scottish (4)	White English (1)	White British (4)		
**Location (first two letters of postcode)**	AB (3)	G (1)	PH (3)	IV (1)	FK (1)
**Marital status**	Divorced (2)	Married (4)	Widowed (1)	Single (1)	With a partner (1)
**Employment status**	Employed (2)	Retired (7)			
**Volunteering experience**	Not a volunteer (2)	Volunteer (7)			

## Data Availability

Due to the autoethnographic nature of this study, data are not available.

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
