# Peer review of "Living and Researching the COVID-19 Pandemic: Autoethnographic Reflections from a Co-Research Team of Older People and Academics"

_ijerph, 2024, doi:10.3390/ijerph21101329_

Round 1

Reviewer 1 Report

Comments and Suggestions for Authors

Article Review 

The article entitled “Living and researching the COVID-19 pandemic: autoethnographic reflections from a co-research team of older people and academics” presents qualitative narratives of “co-researchers” (here I would say “community members”) and 2 academics for the purpose of capturing their experiences as a research team studying older adults in Scotland during the COVID-1 pandemic. The paper is not about the research project itself, but rather a discussion of their collective experiences engaging in the research project. The authors present a clear, concise explanation of the perceived benefits of this kind of work (I call this a PAR project, see below) as well as challenges and some ethical dimensions that arose. It is professionally written, clear in focus, and does a commendable job of using appropriate quotes to drive their insights. This is not a hypothesis testing study, but rather an effort to contribute methodological insights meant to be inform practice and process in health-related social science research. This kind of work is valuable, and I appreciate their collective commitment to sharing their experiences.  

There are two major areas that I would suggest need revision. The first is how the article is framed. The authors use the term “co-reach” or “co-researchers” which they seem to suggest is somewhat novel. They discuss very briefly (lines 73-88) where they see this concept fitting into existing literature. As someone familiar with participatory research (PAR), I would argue this section does not do justice to a deeply historic and now quite common approach to social science research. I would argue that they simply present the perspectives of community partners/members (“co-researchers”) on the project alongside the academics' perspectives (from time to time). I say this because the authors allude to the fact that the co-researchers come from the community – they are referenced as having insider perspectives and assisting in recruiting from within their networks. This makes them community partners. Therefore, the authors need to draw more heavily on the rich literature associated with these research paradigms (PAR, CBPR, Community-Action Research, Action Research, Participatory Research), and it is quite extensive – going back to Freire's pedagogy of the oppressed if not further. I would say this is action oriented because they report the community members felt productive in their endeavors and their work was and would make a difference in their communities. Engaging more extensively in this literature would confirm all their findings since it has already been well documented. For example, PAR gives the community members/co-researchers agency, knowledge (consciousness), purpose, and empowerment etc. Community-members learn new skills, findings are more valid because of the insider (community-partner)/outsider (institutional researcher) perspective, recruitment/buy in from potential participants is higher because of trust and confidence from incorporating community members and dissemination is easier with community-partners. It is highly rewarding, but with many challenges, including mental distress, differences of opinion on what questions to ask or focus on to balance researcher desires/interests/expertise and that of community desires/needs. PAR is also quite time-consuming. We already know this, and the lit review needs to reflect this knowledge.  

This leads to the next substantive concern. If that literature is well covered, as I suggest above, then what, specifically, makes this work novel? Most of what is reported here has already been well documented in PAR/CBPR literature. So why is this important to capture here? What does this contribute? The authors suggest the contribution most clearly in lines 647-653. “This paper adds to the documentation of co-production activity that is currently lacking in health research.” This is simply not true. It is not lacking and a scan of PAR work in the social sciences (health-based or public health-based, in particular) is substantial, especially as decolonizing methodology is becoming standard in most methods classes and in practice. So, the authors, after reviewing this literature, would need to produce from their data a unique contribution to the field/methodology.  

One thing I might suggest is drawing on your narratives that focus on the impact of the broader context on BOTH methodology AND the lived experiences/perceptions of community partners as they engage with PAR during a global pandemic. COVID-19 was, in fact, novel in many ways and did require some substantive adjustments to research methodologies  (you note some of them here as you navigated evolving laws and restrictions in life and in research) to both adjust to the pandemic while capturing, via research, how the pandemic was impacting people (what their project seemed to be about). These insights would be useful to researchers and community members (co-researchers) if another global pandemic were to occur, and if PAR would be something researchers wanted to use during a subsequent pandemic - both of which are highly likely to occur. There are some important lessons learned from conducting this work during a global pandemic that are not likely not already documented in the PAR literatureThe manuscript needs to be re-centered around these contributions in relation to the novelty of a global pandemic. 

Other suggestions: 

  • Define more clearly what is meant by an “ethics of care” approach 

  • What is a “Facts, Feelings and Reflections” review (line 216) 

  • Throughout authors need to make clear who is being quoted – a community researcher/participant or an academic)  

Reviewer 2 Report

Comments and Suggestions for Authors

Brief Summary:

The article presents a collective autoethnography (CAE) of a co-research team that reflects on the experiences of co-researchers during a mixed-methods research project on the lives of older people during the Covid 19 pandemic in Scotland. Results show different ways in which co-researchers not only contributed to in the research process but also benefitted from their work as co-researchers. The article gives an impressive and scientifically sound insight into the experiences of co-researchers in a public health project using collective autoethnography as an appropriate research method. Looking at a participatory research process from the perspective of co-researchers (and their academic colleagues) is a subject on which very little is known. However, since the demands on PPI from funding bodies and the use of participatory research methods become more relevant in health research in recent years, this knowledge is highly relevant from an ethical as well as a methodological point of view.

General concept comments:

Despite the overall good quality of the paper, it could be improved in a small number of points:

·        The introduction section should be shortened substantially and should be written more concisely. (See specific comments below.)

·        The “Materials and Methods” chapter should be restructured. I suggest you provide a short introduction, explaining the different research processes in the original “Healthy Aging in Scotland” (HAGIS) project and the CAE project focussing on the experiences of the co-research team. The description of the research processes itself should be presented in chronological order.

·        The CAE project in itself is a qualitative project. In conventional qualitative projects some precise descriptive information should be presented about the participants forming the sample. This is necessary in order to provide a sound basis for transfering the findings of the project to other settings.
In the CAE project the sample consists of the co-research team (co-researchers and academic researchers). Therefore, it would be helpful to present more precise information on relevant demographic data and other relevant information about all the members of the co-research team in an appropropriate manner (e.g. a table). Some of this information is mentioned in the paper, but only in a general way and it is difficult to get a coherent picture.
However, since not only the academic researchers, but also the co-researchers disclose their full names as authors, this is an ethically complex issue. This issue should be handled with great care to protect the privacy of each author as far as possible and as far as each individual person consents to this. It is imperative that each co-researcher should disclose only such information they explicitly consent to disclose.

·        The aims of the participatory research process in terms of benefits for the co-researchers could be more clearly stated alongside the benefits for the results of the original HAGIS project.

·        There is no doubt that the experiences of the co-researchers reported as results of the CAE project are overwhelmingly positive. However, there are some general remarks in the methods section about “minor crises of confidence” (254) when it came to presenting findings in public, for example. That there are challenges for co-researchers in participative research is well known. The credibility of the findings of the CAE project could be enhanced if, in a kind of “negative case analysis”, not only data on benefits, but also data on some of the challenges experienced by co-researchers were reported.

Specific comments:

line

comment

47f

Please, state more clearly that one of the academic researchers did not participate in the CAE project. Otherwise, it’s a bit confusing, if firstly 3 academics are mentioned (32) and then only 2.

46-58

This information may be presented more concisely and after the discussion about the literature on participatory research or co-research.

59-64

The same questions are mentioned on page 3. Please summarize.

65-100

See above: This review of the literature should be presented before the information about the CAE project.

65-68

Co-production and the work of co-researchers is increasingly prioritized in social science research and there are a growing number of examples of co-research teams developing and undertaking research projects [2-4].”

The references are not sufficient to support the statement about the “growing number of examples”. Please, rephrase in closer relation to the references or add further evidence.

91-93

“Undertaking an egalitarian approach to co-research places demands on co-researchers that require academics to carefully consider the ethics of care, when undertaking co-research [3,10,14].”

The use of the term “co-researchers” is confusing in this context. Writing “… places demands on academic researchers working with co-researchers…” would be more precise.

98

Please, add a methods reference to “collaborative autoethnography”

116f

“The first step was to generate a series of questions to enable us to frame our input to the process which we did during a guided reflective discussion in an online meeting.”

Please, add information about how the decision about the CAE project came about. Who initiated the CAE research process and how was the decision made?

146-259

Please, consider presenting chapter 2. Materials and Methods in chronological order. See general comments above.

268-272

“Co-researchers emphasised their insider status; as people drawn from the same population as the project participants, they share characteristics such as age and generation, and further to this they bring similar life experiences.”

It would be interesting to know how far co-researchers were encouraged to engage in mutual conversations while interviewing or to avoid this, during interview training. It is important to know more about this issue, because it is likely to influence how co-researchers experienced their interviewing.

Which approach to interviewing was adopted: The more distant approach in line with the post-positive research paradigm or a more conversational approach as adopted in research oriented at more constructivist methodologies? (See: Lincoln, Yvonna S.; Lynham, Susan A.; Guba, Egon G. (2018): Paradigmatic Controversies, Contradictions, and Emerging Confluences, Revisited. In: Norman K. Denzin und Yvonna S. Lincoln (Hg.): The SAGE handbook of qualitative research. 5. Aufl. Los Angeles: Sage, S. 108–150.)

This question could, for example, be answered if you provided references about the interview method used and how it was taught during interview training.

279

Please, explain “OCD”.

284

Please, explain “shielder”

324f

“… at times it [the co-research experience] has been emotionally impactful”

Please, explain in more detail in which way this experience was “emotionally impactful”. As mentioned above, it would be helpful to know more about the challenges experienced by co-researchers.

601-603

At the beginning of the discussion, the impact of the research process on co-researchers should also be mentioned. (See general comments above.) In what way did the co-researchers benefit from their participation in the research?

This point is prominent in participatory health research. See, for example: Abma, T., Banks, S., Cook, T., Dias, S., Madsen, W., Springett, J. & Wright, M. T. (2019). Participatory research for health and social well-being. Springer.

656-662

All co-researchers were retired and therefore had much more time available to engage in voluntary work as co-researchers as compared to people who have job duties or family duties to fulfil. The issue of the availabity of time for work as co-researchers is crucial for participatory research. It is highly relevant for the transfer of the findings and even more relevant for other important issues in participatory research, such as recruitment and engagement of co-researchers. Therefore, this aspect should be discussed in more detail.

 Author Response

Round 2

Reviewer 1 Report

Comments and Suggestions for Authors

I do not have any substantive changes or minor suggestions. There are typos throughout the manuscript. There needs to be close editing. I noted them but there are too many to report ranging from missing commas to run-on sentences. 

In one of the additions made (lines 91-93) there is a stand along paragraph that is only one sentence long. This sentences says there were creative shifts in methodology during the pandemic. Even though their paper discusses their shifts, in this section it would be helpful to delineate how social scientists did shift by providing specific examples and why they made those methodological changes. It may seem intuitive, but there are many scholars who would benefit from hearing about those changes. 

Finally, the authors seem to want to center the experiences of the non-academic participants. This is really the purpose of participatory methods - to decolonize the production of knowledge. They state this. My only advice would be to expand on how this was/is emancipatory. For example, learning new skills, engaging with new technologies, presenting at conferences - these are tangible and pragmatic skills that extend beyond the life of the project and are worth highlighting a bit more. They talk about the co-production of knowledge, which is also  essential to highlight. I think they do this part well. This is redistributing power and why scholars do this work. So just a bit more emphasis on this in the discussion would be fruitful. 

Author Response

I do not have any substantive changes or minor suggestions. There are typos throughout the manuscript. There needs to be close editing. I noted them but there are too many to report ranging from missing commas to run-on sentences. 

I have completed a careful proof read and used the available tools on Microsoft Word to correct grammar and improve clarity. I have not adopted the Oxford comma rule but can do if required by the journal.

In one of the additions made (lines 91-93) there is a stand along paragraph that is only one sentence long. This sentences says there were creative shifts in methodology during the pandemic. Even though their paper discusses their shifts, in this section it would be helpful to delineate how social scientists did shift by providing specific examples and why they made those methodological changes. It may seem intuitive, but there are many scholars who would benefit from hearing about those changes. 

Changes are highlighted in yellow providing more detail on new methods and approaches adopted during the pandemic.

Finally, the authors seem to want to center the experiences of the non-academic participants. This is really the purpose of participatory methods - to decolonize the production of knowledge. They state this. My only advice would be to expand on how this was/is emancipatory. For example, learning new skills, engaging with new technologies, presenting at conferences - these are tangible and pragmatic skills that extend beyond the life of the project and are worth highlighting a bit more. They talk about the co-production of knowledge, which is also  essential to highlight. I think they do this part well. This is redistributing power and why scholars do this work. So just a bit more emphasis on this in the discussion would be fruitful. 

Additional text has been added to the discussion to better emphasise these points, text is highlighted in yellow.